# Molecular organization and dynamics of the fusion protein Gc at the hantavirus surface

Eduardo A Bignon[1], Amelina Albornoz[1], Pablo Guardado-Calvo[2], Félix A Rey[2]*, Nicole D Tischler[1]*

[1]Laboratorio de Virología Molecular, Fundación Ciencia & Vida, Santiago, Chile; [2]Structural Virology Unit, Virology Department, Institut Pasteur, CNRS UMR 3569, Paris, France

**Abstract** The hantavirus envelope glycoproteins Gn and Gc mediate virion assembly and cell entry, with Gc driving fusion of viral and endosomal membranes. Although the X-ray structures and overall arrangement of Gn and Gc on the hantavirus spikes are known, their detailed interactions are not. Here we show that the lateral contacts between spikes are mediated by the same 2-fold contacts observed in Gc crystals at neutral pH, allowing the engineering of disulfide bonds to cross-link spikes. Disrupting the observed dimer interface affects particle assembly and overall spike stability. We further show that the spikes display a temperature-dependent dynamic behavior at neutral pH, alternating between 'open' and 'closed' forms. We show that the open form exposes the Gc fusion loops but is off-pathway for productive Gc-induced membrane fusion and cell entry. These data also provide crucial new insights for the design of optimized Gn/Gc immunogens to elicit protective immune responses.

DOI: https://doi.org/10.7554/eLife.46028.001

## Introduction

Hantaviruses (order *Bunyavirales*, family *Hantaviridae*, genus *Orthohantavirus*) persistently infect rodents throughout the world. When transmitted to humans they can cause serious disease such as hemorrhagic fever with renal syndrome and hantavirus pulmonary syndrome (HPS) with case fatalities up to 10% and 40%, respectively (*Krüger et al., 2015*). Yet there are no efficient preventive nor therapeutic measures approved against these diseases. As other members of the *Bunyavirales* order, their RNA genome is single stranded with negative polarity, composed of three segments. The medium (M) segment encodes a membrane-anchored polyprotein precursor GPC, which is processed by host cell signal peptidases to generate glyoproteins Gn and Gc (*Löber et al., 2001*). Virion-like particles (VLPs) are formed when GPC is expressed in the absence of other viral proteins (*Acuña et al., 2014*), indicating an important role of the glycoproteins in virion budding and in cell exit of the progeny. Authentic virions and VLPs have been shown to project spikes organized in a square lattice (*Acuña et al., 2014*; *Martin et al., 1985*). The work of *Hepojoki et al. (2010)* revealed that Gn/Gc species can be covalently crosslinked on the surface of virions and suggested oligomeric models for spike assembly based on the characterization of detergent-solubilized spikes. Electron cryo-tomography (cryo-ET) data revealed spikes with volumes that can accommodate the molecular mass of (Gn/Gc)$_4$ hetero-octamers, related by 2-fold symmetry axes oriented radially in the particle (*Battisti et al., 2011*; *Huiskonen et al., 2010*). A higher resolution 15.6 Å cryo-ET map allowed the docking of the Gn ectodomain into the central lobes on the tetrameric spike, at the membrane distal side, and masking the Gc fusion loops, suggesting that the 2-fold related spike-spike interactions are made by the Gc moiety (*Li et al., 2016*). Moreover, a central role of Gn in self-association to

*For correspondence:
felix.rey@pasteur.fr (FéAR);
ntischler@cienciavida.org (NDT)

**eLife digest** Hantaviruses infect rodents and other small mammals, but do not harm them. When transmitted to humans, often through rodent urine, feces or saliva, they can cause serious and even fatal diseases. Currently, there are no known methods that effectively prevent hantavirus infections or treat the diseases that they cause.

During an infection, viruses invade the cells of their host. A hantavirus interacts with target cells through proteins on its surface called Gn and Gc glycoproteins. Previous work has shown that these glycoproteins are organized in bundles of four Gn and four Gc proteins, termed spikes, which project from the membrane that surrounds the virus. The Gc protein changes shape when it is activated and exposes a hidden region that can insert into the membrane of the target cell. The Gc proteins then change shape again to force the cell to fuse with the viral membrane. This process allows the virus to be taken up into the cell, where it can replicate.

While the structures of each viral glycoprotein have been determined in isolation, it was not known how they interact within the Gn/Gc spike. Such information is crucial to understand how the viruses infect cells and which areas are exposed to the immune system of the host – and so could be targeted by antiviral treatments.

Bignon et al. have now identified the molecular contacts that occur between spikes and interconnect them into a grid-like lattice on the surface of the virus. Genetically altering specific sections of the Gc glycoprotein strengthened or weakened these contacts, which correspondingly increased or decreased how stable the spike was. Preventing the contacts from forming resulted in cells releasing fewer virus-like particles.

Bignon et al. also show that at the body temperature of mammals, the shape of the spike fluctuates between an 'open' form that exposes the region of Gc that inserts into the cell membrane, and a 'closed' form that hides this region. However, when Gc is activated, the open form becomes unable to cause the viral and cell membranes to fuse together.

Together, the results presented by Bignon et al. help us to understand how changes to the hantavirus surface enable the virus to infect cells. This knowledge will help researchers to design vaccines that protect against hantavirus infections.

DOI: https://doi.org/10.7554/eLife.46028.002

form spikes has recently been confirmed by number and brightness analysis in single live cells showing that separate Gn expression allows detection of Gn oligomers while separate Gc expression predominantly leads to Gc monomers and some Gc dimers (*Sperber et al., 2019*).

The Gn/Gc spikes on the viral surface are key in directing entry into new cells (*Cifuentes-Muñoz et al., 2014*). Hantavirus cell entry occurs by the interaction of the envelope glycoproteins with host cell receptors, which leads to viral uptake into endosomes. Cell entry is completed when Gc induces the fusion of the viral envelope with the endosomal membrane at acidic pH (*Acuña et al., 2015*). Structure-function studies have also confirmed that Gc is a class II fusion protein, and have provided insight into its fusogenic conformational change triggered by low pH (*Barriga et al., 2016*; *Guardado-Calvo et al., 2016*; *Tischler et al., 2005*; *Willensky et al., 2016*). This irreversible structural rearrangement of Gc into a stable post-fusion trimer involves several steps, including the initial exposure of the Gc fusion loops, which then insert into the target cell membrane via an extended trimeric intermediate. The individual Gc trimer subunits then adopt a 'hairpin' conformation that forces apposition of viral and cellular membranes, to allow the bilayers to fuse. On the virion, the Gn and Gc residues involved in intra- and inter-spike interactions have not been identified. These interactions control the fusion activity of Gc (*Guardado-Calvo and Rey, 2017*), by maintaining it at neutral pH in a functional metastable conformation (*Harrison, 2015*).

Here, we show that the 2-fold Gc:Gc contacts between adjacent $(Gn/Gc)_4$ spikes at the surface of the hantavirus particles are mediated by the same interface observed in a crystallographic dimer revealed by the available X-ray structure of Gc in a pre-fusion form. These contacts regulate viral assembly together with spike stability and subsequent disassembly for entry. We further demonstrate that at physiological temperature, the spikes exhibit a dynamic temperature-dependent

equilibrium between a 'closed' form in which the fusion peptides are masked by Gn, and an 'open' form that allows the particle to bind to liposomes at neutral pH.

## Results

### Engineering of an inter-spike disulfide bond based on the crystallographic Gc dimer

The X-ray structure of the Hantaan virus Gc ectodomain at neutral pH (PDB: 5LJY) (*Guardado-Calvo et al., 2016*) revealed a pre-fusion monomer which, among other crystal contacts, presented two Gc molecules packing about a crystallographic 2-fold axis. The two Gc monomers in this dimer cross at an angle of roughly 50 degrees (*Figure 1a*) similar to the 2-fold related spike interactions in the cryo-ET map (*Li et al., 2016*). The 'dimer' interaction observed in the hantavirus Gc crystals is reminiscent of the crystallographic dimer contacts presented by class II alphavirus fusion protein E1 (*Roussel et al., 2006*), which is recapitulated by the 2-fold related contacts between hetero-hexameric (E2/E1)$_3$ spikes at the surface of alphavirus particles (*Sun et al., 2013*; *Voss et al., 2010*) (*Figure 1a*).

To test whether the Gc:Gc interface observed in the crystallographic dimer is involved in contacts between adjacent spikes at the surface of hantavirus particles, we introduced cysteine substitutions of candidate Gc residues at the putative 2-fold interface, such that they could form inter-spike disulfide bonds. We surveyed residues facing each other across the crystallographic Gc dimer interface with Cα-Cα distances ranging between 4 and 10 Å and selected the highly conserved His303 and Gly187 for single cysteine substitutions. These residues face their counterpart on the 2-fold axis of the crystallographic dimer with Cα-Cα distances of 8.4 Å (His303-His303) and 5.6 Å (Gly187-Gly187) (*Figure 1b*). Because the Gc:Gc interface residues are highly conserved (*Figure 1—figure supplement 1*), the observed contacts made by Hantaan virus Gc should be maintained in all hantaviruses. We therefore turned to an Andes virus (ANDV) glycoprotein expression and VLP producer system for the functional experiments (*Acuña et al., 2014*), which has the advantage of corroboration of ANDV VLP data with authentic ANDV, which we can manipulate. We tested ANDV mutant Gn/Gc constructs having either the Gc H303C or the G187C substitutions (the amino acid sequence numbering is the same between Andes and Hantaan virus Gc) for protein production in 293FT cells, surveyed their transport to the plasma membrane as a measure for proper protein folding, and monitored their assembly into VLPs released in the cells' supernatant. We found that the cysteine mutants were properly expressed and trafficked to the plasma membrane (*Figure 1c*). We next analyzed the presence of VLPs in the concentrated cells' supernatant by reducing and non-reducing SDS polyacrylamide gel electrophoresis (PAGE) and western blot. The wild type Gc migrated as a monomer under both conditions, while the Gc cysteine mutants migrated predominantly with a molecular mass of ~100 kDa, corresponding to Gc dimers, under non-reducing conditions. In the presence of a reducing agent, these Gc dimers were readily dissociated to monomers (*Figure 1d*). Together, these results confirm, in a biological context, that the residues forming the Gc dimer contacts in the X-ray structure of a pre-fusion form of Gc are proximal enough to each other on viral particles to allow for disulfide formation while still forming VLPs, thereby supporting the biological relevance of the crystallographic Gc dimer. Furthermore, when comparing the yields of VLP production, the G187C mutant resulted in higher production levels than H303C (*Figure 1d*). This result correlates with the better geometry and distances between the Cα atoms in the crystallographic Gc dimer for G187C compared to H303C (*Figure 1b*).

We compared the spikes of the G187C mutant VLPs to the wild type VLPs in terms of their migration profile in blue native PAGE (BN-PAGE) combined with western blotting upon detergent-solubilization of the spikes. When we incubated Andes VLPs bearing wild type (Gn/Gc)$_4$ spikes at 20°C and neutral pH, the detergent-solubilized spike was identified as a single band recognized by both, anti-Gn and anti-Gc MAbs (*Figure 2a*). This band, corresponding to a molecular weight of ~500 kDa matching a (Gn/Gc)$_4$ spike, migrated roughly as expected in BN-PAGE, given the migration of the individual Gn and Gc monomers (see migration at 50°C), of the Gc post-fusion homotrimer (see migration at acidic pH), and of the standard reference bands at 480 and 720 kDa. In contrast, in the detergent-solubilized G187C VLPs the band containing both, Gn and Gc, barely entered the gel at

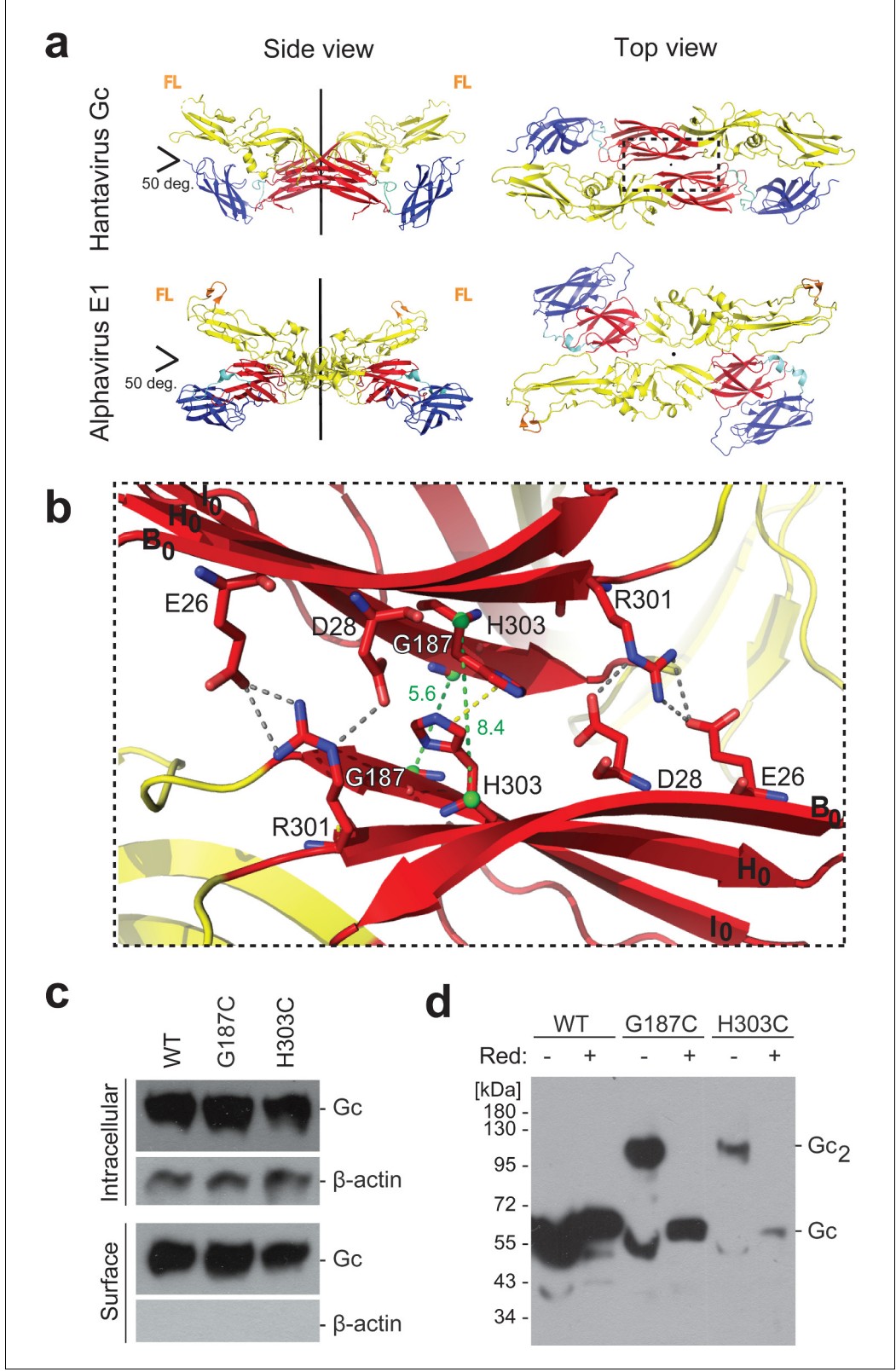

**Figure 1.** A cysteine residue engineered at the crystallographic Gc dimer interface cross-links spikes on viral particles. (a) X-ray structure of the crystallographic Hantaan virus Gc dimer (upper panel, PDB: 5LJY, *Guardado-Calvo et al., 2016*), displayed alongside the crystallographic alphavirus E1 dimer (PDB:2ALA, *Roussel et al., 2006*) also observed in the contact between spikes on the alphavirus particles (PDB:3J2W, *Sun et al., 2013*) (lower

*Figure 1 continued on next page*

*Figure 1 continued*

panel). The two-fold axis is drawn in black. The three domains are labeled; domain I, red; domain II, yellow; domain III, blue. The fusion loops are indicated in orange (FL). Domain III of Hantaan virus Gc appears to have adopted an orientation different than would be expected in the spike, but as it is not part of the dimer interface, it does not affect the contacts made by domain I examined here. (b) Closeup of the crystallographic Gc dimer interface, view slightly tilted with respect to the 2-fold axes to show the selected residues for cysteine substitution as well as other residues at the interface, shown in sticks color-coded according to atom type (nitrogen blue, oxygen red, carbon same as the domain color). The Cα of Gly187 and H303 are as green spheres and the Cα-Cα distances across the interface for Gly187-Gly187 and His303-His303 are drawn as green dotted lines. Salt bridges and hydrogen bonds between the carboxyl groups of Glu26 and Asp28 with the guanidinium group of Arg301 are drawn as gray dotted lines. The imidazole rings of His303 interact by π-stacking, indicated with a yellow dotted line. The domain I β-strands $B_0$, $H_0$ and $I_0$ from each protomer are labeled. (c) Expression yields and cell localization of ANDV Gc mutants G187C and H303C representative for two biological replicates. Western blot of fractions obtained from 293FT cells expressing Gn and wild type (WT) or mutant Gc after surface biotinylation using anti-Gc or anti-β-actin antibodies. (d) SDS Page and western blot under reducing and non-reducing conditions of VLPs obtained from supernatants of cells expressing wild type Gn together with wild type, G187C or H303C Gc, representative for four biological replicates. The absence or presence of the reducing agent β-mercaptoethanol is indicated by Red: - and +, respectively.
DOI: https://doi.org/10.7554/eLife.46028.003

The following figure supplement is available for figure 1:

**Figure supplement 1.** Conservation of Gc:Gc dimer interface contacts.
DOI: https://doi.org/10.7554/eLife.46028.004

temperatures up to 40°C, indicating a high molecular mass, as expected if a disulfide bond interconnects multiple adjacent spikes (*Figure 2b*).

## The inter-spike Gc dimer contacts are required for VLP assembly

The above data on the disulfide bonds suggested that the dimer interface observed in the Hantaan virus Gc crystals is indeed involved in lateral interactions between spikes on hantavirus particles. The crystal contacts at this interface bury a surface area of ~545 Å$^2$ per subunit, which is a relatively small contact patch (*Figure 1b*), consistent with the requirement for particle dissociation for entry into cells. We noted that this interface contains several strictly conserved polar residues, three charged (Glu26, Asp28, Arg301), and one ionizable (His303) (*Figure 1—figure supplement 1*), which make a network of hydrogen bonds, including inter-chain salt bridges, as well as π-stacking of the His303 imidazole rings (*Figure 1b*).

To further assess the potential functional relevance of this interface, we introduced the following individual site-directed mutations in Gc: E26A, D28A, R301A and H303A. As with the cysteine mutants described above, we tested the new mutant ANDV Gn/Gc constructs for glycoprotein production in 293FT cells, transport to the plasma membrane and assembly into VLPs with concomitant release from cells (*Figure 3a* and *Figure 3—figure supplement 1a and 1d*). Of the single Ala substitutions, only mutant D28A passed all the above tests, implying that this mutation was well tolerated, albeit yielding significantly reduced amounts of VLP release (*Figure 3a*). The three other mutants were either not detected by western blot (E26A) or led to the expression of truncated versions of the protein (30 kDa) in 293FT and Vero E6 producer cells (R301A and H303A) (*Figure 3a* and *Figure 3—figure supplement 1a and 1e*). Yet, substitution of these residues by more chemically similar amino acids, such as the E26Q, R301Q and H303F mutants, still allowing interactions across the interface, resulted in their detectable expression, transport to the plasma membrane and equivalent amounts of assembly into VLPs (*Figure 3a* and *Figure 3—figure supplement 1b*). To introduce repulsion at the Gc:Gc interface, we also replaced the residues with opposite charges: Gc E26K, D28K, R301E and H303E. Although these mutants could be detected in the intracellular and plasma membrane fractions, their release into the supernatant was significantly decreased (*Figure 3a* and *Figure 3—figure supplement 1c*), suggesting that mutations interfering with interface interactions, such as alanine and opposed charges, strongly impair VLP formation. Taken together, these results indicate that the observed 2-fold related Gc:Gc inter-chain contacts are crucial for viral particle assembly, as would be expected if this interface were indeed the site of lateral packing between adjacent hantaviral spikes.

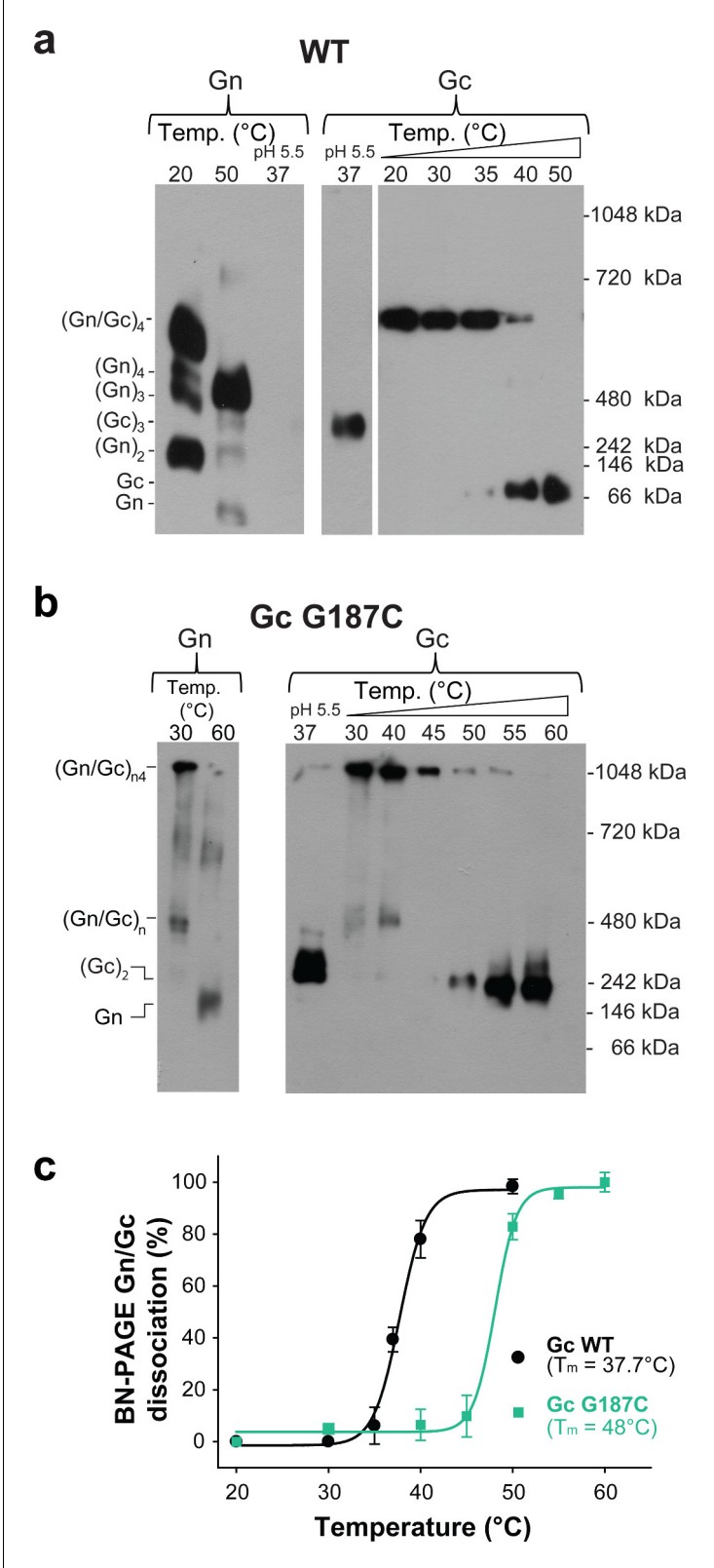

**Figure 2.** The covalent Gc:Gc dimer disulfide bond G187C increases Gn/Gc spike stability. (**a–b**) Representative BN-PAGE and western blotting of ANDV wild type (**a**) and G187C mutant spikes (**b**). The spikes were extracted from VLPs by Triton X-100 and treated at the indicated temperatures of 20–60°C at neutral pH. The presence of Gn or Gc in each lane was detected by western blot analysis by splitting the transferred gel in two parts and revealing

*Figure 2 continued on next page*

*Figure 2 continued*

with anti-Gn (left panel) and anti-Gc (right panel) antibodies. As internal control for Gc species migration, Gc wild type homotrimers were examined in each gel by treatment of VLPs at pH 5.5. No signal for Gn was detected when treated at low pH, suggesting that either the mAb may not react with Gn in native gels when forming a more compact tetramer (*Rissanen et al., 2017*), or that Gn may not enter the native gels. To further estimate the oligomerization species of Gn and Gc (indicated on the left side of the blot), the migration of their monomeric and multimeric forms was compared with a native protein standard (indicated on the right side of the blot). (c) Graph of the temperature-induced Gn/Gc dissociation of detergent solubilized spikes from wild type or G187C mutant VLPs and quantified by densitometry. Averages ± s.d. from three biological replicates are shown. The curves were fitted using a sigmoidal equation (*Equation 1*) and are indicated as a line.

DOI: https://doi.org/10.7554/eLife.46028.005

The following source data is available for figure 2:

**Source data 1.** Original blots and data points for *Figure 2c*.
DOI: https://doi.org/10.7554/eLife.46028.006

## The dissociation temperature of the detergent-solubilized spikes is affected by mutations at the Gc:Gc dimer interface

We used BN-PAGE to compare the stability at increasing temperatures of the detergent-solubilized hantavirus wild type and mutant spike complexes. When we treated the wild type spike complex at different temperatures up to 50°C, we were able to visualize on the gel that the band corresponding

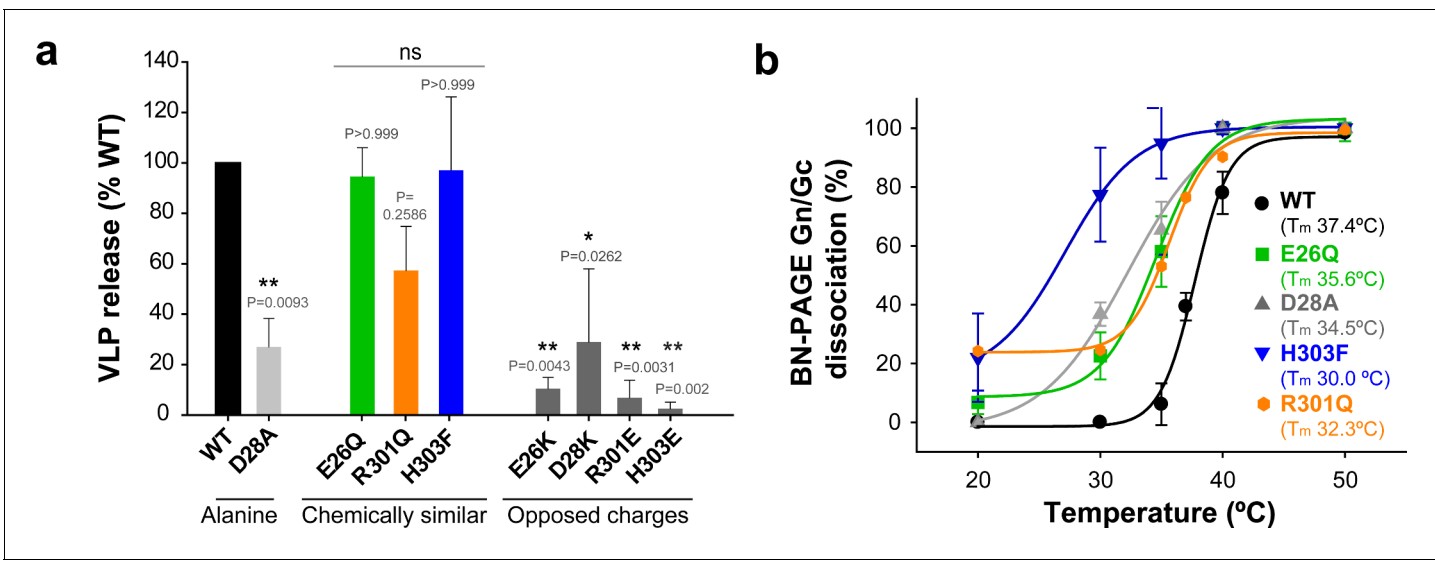

**Figure 3.** The inter-spike Gc:Gc dimer interface affect VLP release and hantavirus Gn/Gc spike stability. a) VLP assembly and release of ANDV Gc mutants including Ala-substitutions, chemically similar mutations and opposed charge substitutions quantified from western blot analysis using anti-Gc antibody of concentrated supernatant (VLP) obtained from 293FT cells expressing Gn and wild type or mutant Gc. Averages ± s.d. representative for two biological replicates are shown relative to VLP wild type (WT) release. Data were analyzed by one-way ANOVA with a Bonferroni adjustment for multiple comparisons, p<0.001 (***), p<0.01 (**), p<0.05 (*). b) Graph of the temperature-induced Gn/Gc dissociation of the different Gc mutants compared to wild type quantified by densitometry. Averages ± s.d. from two biological replicates are shown. All dissociation curves were fitted using a sigmoidal equation (*Equation 1*) and are indicated as a continuous or dotted line.

DOI: https://doi.org/10.7554/eLife.46028.007

The following source data and figure supplements are available for figure 3:

**Source data 1.** Data points for graphs of *Figure 3a and b*.
DOI: https://doi.org/10.7554/eLife.46028.008

**Figure supplement 1.** Characterization of the expression yields, trafficking and VLP assembly of ANDV Gc mutants.
DOI: https://doi.org/10.7554/eLife.46028.009

**Figure supplement 2.** Thermal stability of the Gn/Gc spike complex from Gc:Gc interface mutants.
DOI: https://doi.org/10.7554/eLife.46028.010

to the (Gn/Gc)$_4$ spikes at 20°C and 30°C gradually disappeared at higher temperatures with a concomitant appearance of smaller migration bands, which corresponded to several oligomeric Gn forms and to a monomeric Gc species (*Figure 2a*). The absence of intermediate Gn/Gc dissociation products suggested a two-state behavior, in a highly cooperative spike dissociation process. Quantification of the temperature-induced dissociation of the detergent-solubilized wild type ANDV Gn/Gc spike revealed a melting temperature (T$_m$) of 37.7 ± 0.4°C. In comparison, the Gn/Gc band of the G187C VLPs did not dissociate into homooligomeric Gn or Gc species up to temperatures of 45°C, revealing a 10°C increase of its T$_m$ to 48°C (*Figure 2b*). The dissociated Gc species migrated to a further distance than the band corresponding to a wild type Gc homotrimer (see *Figure 2a*, right panel), in line with the expected migration of a disulfide-linked Gc dimer (*Figure 2b*). With the other Gc dimer interface mutants, we found a T$_m$ decreased by 2–5°C for the mutants E26Q, D28A, R301Q - each affecting ionic interactions at the interface (*Figure 3d* and *Figure 3—figure supplement 2*). In the case of the H303F mutant, the T$_m$ dropped by 7°C. This more important effect in the thermostability of H303F may be explained by the fact that the phenylalanine side chain is bulkier than that of histidine, hence forcing the Gc protomers to re-accommodate to the change and affect the overall Gc:Gc interaction. Together, these data suggest that the intra-spike Gn/Gc interactions are affected by the lateral inter-spike Gc:Gc contact by allostery; when the Gc:Gc contacts are strengthened by a disulfide bond the T$_m$ of the Gn/Gc spike raise, and lower the spike T$_m$ when the Gc:Gc contacts are weakened.

## Residues at the inter-spike Gc:Gc contacts affect the pH-triggered fusion activation

We also monitored the effect of mutations at the Gc:Gc inter-spike interface on the pH required for spike activation for membrane fusion. Among all Gc:Gc interface mutants, only Gc E26Q induced syncytia formation of cells expressing the mutant Gn/Gc construct upon incubation at pH 5.5,

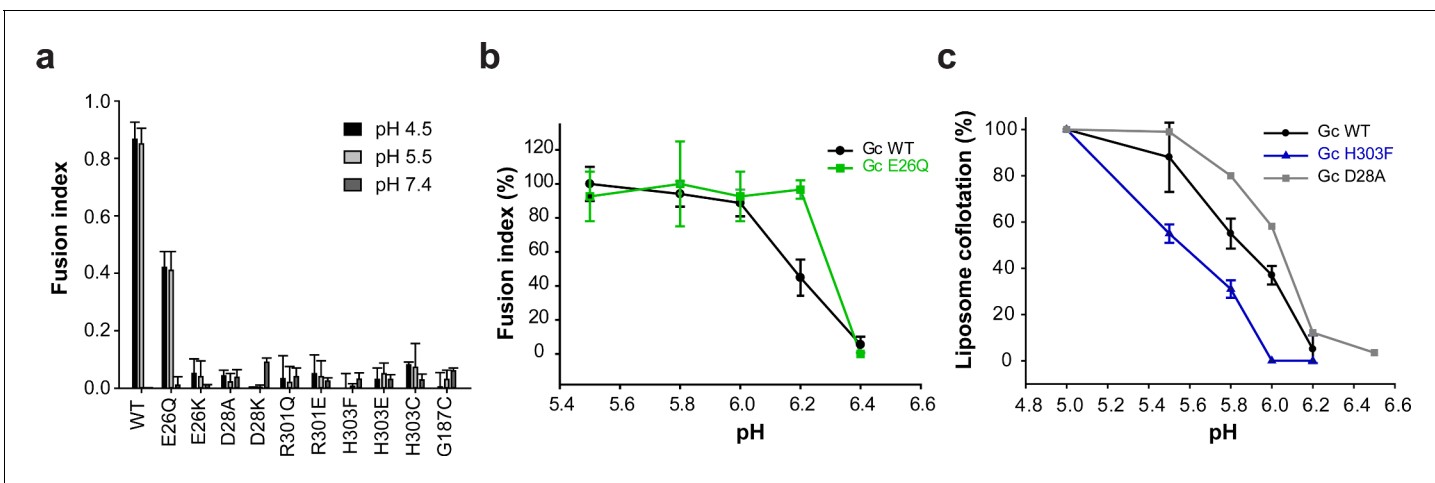

**Figure 4.** The Gc:Gc dimer contacts modulate low pH fusion activation. (**a-b**) Fusion activity of cells expressing ANDV wild type Gn/Gc (WT) or wild type Gn/mutant Gc at different pHs. Syncytia formation was induced by lowering the pH to 5.5 or 4.5 and was quantified by counting cells and nuclei using three color fluorescence microscopy. Averages ± s.d. from three biological replicates are shown. In (**b**) the maximal fusion activity of WT or mutant Gn/Gc was normalized by its setting to 100% in each case. (**c**) Liposome coflotation assay to visualize acid-induced activation and membrane interaction of WT and mutant VLPs. VLPs were incubated with DPH-labeled liposomes at different pHs at 37°C. Fractions of the step gradient sedimentation were examined for the presence of Gc by western blot and liposomes by fluorescence. Western blots were quantified by densitometry and averages ± s.d. from two (D28A) and three (WT and H303F) biological replicates are shown.

DOI: https://doi.org/10.7554/eLife.46028.011

The following source data and figure supplement are available for figure 4:

**Source data 1.** Data points for graphs of *Figure 4a, b and c*.

DOI: https://doi.org/10.7554/eLife.46028.013

**Figure supplement 1.** Acid-induced activation and membrane interaction of WT and mutant VLPs assessed by a VLP-liposome coflotation assay.

DOI: https://doi.org/10.7554/eLife.46028.012

retaining ~50% of the fusion activity of the wild type (*Figure 4a*). All other mutants were fusion inactive, even at a pH as low as 4.5. When we assayed the pH for triggering cell-cell fusion by the E26Q mutant we observed an activation pH of 6.2, that is 0.2 units higher than wild type (*Figure 4b*). Such a raise in the activation pH may be explained by the loss of a salt bridge between Glu26 and Arg301 in the Gc:Gc inter-spike dimer (*Figure 1b*), facilitating spike dissociation and therefore leading to fusion at a less acidic pH compared to wild type.

In order to monitor the activation step allowing interaction with membranes of wild type and Gc: Gc interface mutant VLPs, we carried out liposome co-flotation studies at pH values ranging from 5.0 to 6.4. For this purpose, we mixed fluorescently labeled liposomes with the VLPs at each pH, and loaded the mixture to the bottom of a sucrose step gradient. After centrifugation, we monitored each fraction for the presence of liposomes (by fluorescence) and VLPs (by western blot against Gc). At pH 6.2, the liposomes migrated to the top of the gradient while the wild type VLPs remained in the bottom fractions (*Figure 4c* and *Figure 4—figure supplement 1a*), but increasing amounts of the VLPs were observed in the top fractions at more acidic pHs, depending on the mutant. The D28A mutant began to float at pH 6.2, 0.2 units higher compared to wild type VLPs (*Figure 4c* and *Figure 4—figure supplement 1b*). Contrary to E26Q, mutant D28A is inactive in syncytia formation. The X-ray structures show that Asp28 not only contributes to the Gc dimer interface (*Figure 1b*), but its side chain is also involved in a network of inter-subunit polar interactions stabilizing the postfusion Gc trimer (*Guardado-Calvo et al., 2016*). Again, in this mutant, like in E26Q, the destabilization of the Gc dimer interaction caused fusion activation at a less acidic pH. And the destabilization of the post-fusion D28A Gc trimer likely renders it incompetent for inducing fusion, unlike E26Q.

When we analyzed the H303F mutant for pH-induced liposome coflotation, we found that it was more resistant to activation and required a 0.2 units lower pH for fusion activation compared to wild type (*Figure 4c* and *Figure 4—figure supplement 1c*), opposite to the effect of the E26Q and D28A mutants (*Figure 4b–c*). Hence, although this mutation led to considerable destabilization of the spike in terms of its thermal resistance (*Figure 3d*), it turned out to be more resistant to acidification (*Figure 4c*). Taking into account that the His303 imidazole rings face each other across the interface with a distance of 4.1 Å (*Figure 1b*), they very likely undergo a strong electrostatic repulsion upon protonation at acidic pH. Thus, when His303 is substituted by phenylalanine, this effect does not occur, and additional residues elsewhere must become protonated in order to trigger spike dissociation and fusion. Given that the H303F mutant was fusion inactive at any tested pH, the His303 role in fusion remains to be understood. Together, from these data we conclude that the hantavirus lateral inter-spike Gc:Gc interactions indirectly control spike stability, influencing at the same time Gn/Gc dissociation to induce membrane fusion.

## Temperature-induced Gc fusion loop exposure on VLPs at neutral pH

Previous data on the hantavirus spike organization (*Li et al., 2016*) – and that of other class II enveloped viruses such as alphaviruses and phleboviruses (*Guardado-Calvo and Rey, 2017*; *Halldorsson et al., 2018*; *Voss et al., 2010*) – suggest that Gn conceals the Gc fusion loops at the top of the spikes, keeping them from premature membrane insertion until exposure to low pH. We hypothesized that the temperature-induced Gn/Gc dissociation observed by BN-PAGE could reflect a conformational change within the spike, which would lead to a looser interaction between Gn and Gc at the VLP surface. The lateral inter-spike interaction - absent in the detergent-solubilized spikes - may restrain full dissociation of Gn and Gc on VLPs, and the Gn/Gc dissociation observed by BN-PAGE may reflect a temperature-induced transition of the spike into a state in which the fusion loops become exposed at the top of the hantavirus surface at neutral pH. To test this notion, we incubated wild type Andes VLPs with liposomes at pH 7.4 at different temperatures and assayed them in the VLP/liposome coflotation assay. When the incubation was performed at temperatures in the range from 20°C to 30°C, we found the fluorescence-labeled liposomes at the top of the gradient while the VLPs remained in the bottom fractions (*Figure 5a*), confirming previous data that VLPs and liposomes do not interact under these conditions (*Acuña et al., 2015*). However, when increasing the temperature to 37°C and above, we observed that at neutral pH ANDV particles floated with liposomes to the upper fractions, increasing gradually with temperature (*Figure 5a*). To assess whether membrane interaction at temperatures ≥ 37°C was specifically induced by the Gc fusion loops, and not by non-specific interactions, we tested liposome coflotation of Andes VLPs bearing the W115A/ F250A mutations in Gc. These two substitutions of aromatic residues to alanine, respectively at the

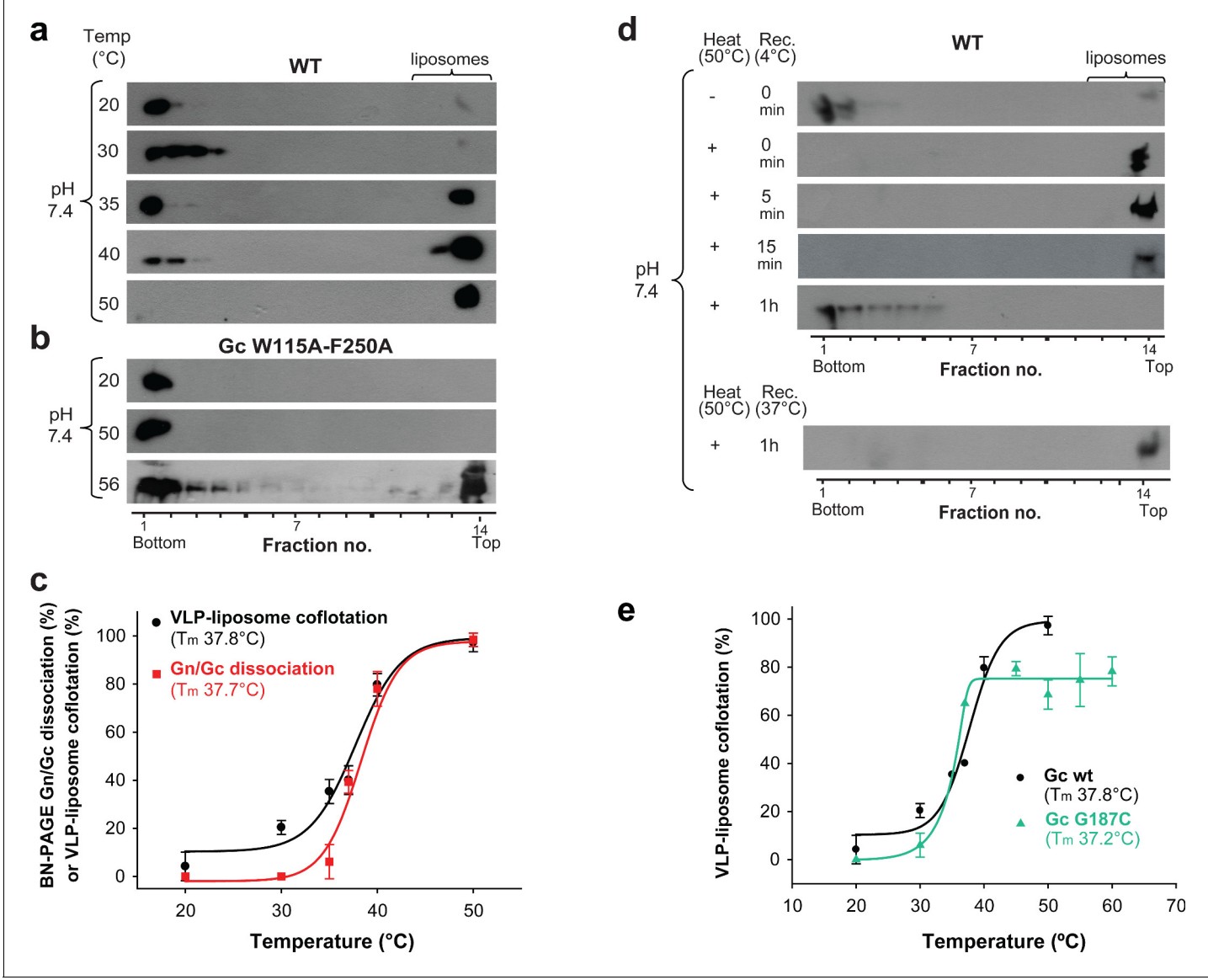

**Figure 5.** Temperature-induced Gc fusion loop exposure. (a-b) Liposome coflotation assay to visualize temperature-induced membrane interaction of ANDV wild type (WT) VLPs (a) or ANDV VLPs bearing the Gc fusion loop mutant W115A/F250A (b). VLPs were incubated with DPH-labeled liposomes at pH 7.4 at the indicated temperatures. After flotation in a step gradient, fractions were examined for the presence of liposomes by fluorescence and Gc by western blot. (c) Graph indicating temperature-induced Gn/Gc spike dissociation by BN-PAGE (from *Figure 2a*) and temperature-induced wild type VLP-liposome coflotation (from *Figure 5a*) quantified by densitometry. Averages ± s.d. from three biological replicates are shown. Both curves (indicated as a line) were fitted using sigmoidal *Equations 1 and 2*, respectively, and show a 50% response at approximately ~37˚C. $T_m$ indicates either the 50% Gn/Gc dissociation temperature of Triton X-100 solubilized spikes or the temperature at which 50% of the spikes expose the fusion loops allowing for liposome coflotation. (d) Liposome coflotation assay showing the reversibility of the fusion loop exposure. VLPs were incubated at 50˚C for 15 min. Then the heat-treated VLPs were allowed to recover at the indicated temperature and time, and next incubated for 5 min with liposomes at RT. Flotation and detection was performed as in (a-b). Rec., recovery. Result representative for three biological replicates. (e) Graph of the temperature-induced flotation of liposome with VLPs bearing the Gc disulfide mutant G187C compared to wild type (WT) quantified by densitometry. Averages ± s.d. from three biological replicates are indicated. The fitting of the curves was performed by using a sigmoidal equation (*Equation 2*). $T_m$ indicates the temperature at which 50% of the spikes expose the fusion loops inducing membrane interaction.

DOI: https://doi.org/10.7554/eLife.46028.014

The following source data and figure supplement are available for figure 5:

**Source data 1.** Data points for graphs of *Figure 5c and d*.

DOI: https://doi.org/10.7554/eLife.46028.016

*Figure 5 continued on next page*

*Figure 5 continued*

**Figure supplement 1.** Temperature-induced Gc fusion loop exposure of VLPs bearing the Gc 187C mutant measured by a VLP-liposome coflotation assay.

DOI: https://doi.org/10.7554/eLife.46028.015

tip of the *cd* and *ij* loops (which are two of the three fusion loops of Gc), do not interfere with VLP formation but were previously shown to abolish insertion into target membranes at low pH (*Guardado-Calvo et al., 2016*). In contrast to wild type, high temperature treatment of the fusion loop mutant VLPs up to 50°C at pH 7.4 did not lead to flotation with liposomes to the upper fractions (*Figure 5b*), indicating that both, the acid-pH-induced and the temperature-induced interaction of VLPs with liposomes, were specifically driven by the Gc fusion loops. In contrast, treatment at 56°C resulted in flotation of the fusion loop mutant with the liposomes, indicative of non-specific interactions with membranes likely due to partial protein denaturation and concomitant exposure of hydrophobic regions (*Figure 5b*).

Comparison of the temperature-induced Andes VLP-liposome interaction at neutral pH (*Figure 5a*) with the temperature-induced dissociation of detergent-solubilized ANDV spikes (*Figure 2a*), showed the same profile (*Figure 5c*). It revealed a $T_m$ for the conformational transition - defined as the temperature at which 50% of the VLPs floated with the liposomes - of 37.8 ± 1.1°C, matching the $T_m$ of 37.7 ± 0.4°C for the detergent-solubilized spike dissociation. Both ANDV Gn/Gc dissociation and fusion loop exposure as a function of temperature followed a sigmoidal curve, indicative of a two-states system.

## The temperature-induced 'closed' to 'open' spike transition is reversible

To test the reversibility of the observed transition, we incubated the VLPs for 15 min at 50°C to induce fusion loop exposure at neutral pH, and then back-treated them for 1 h at 4°C to see if the spikes would return to their initial, 'closed' conformation. When we then assayed these VLPs in the liposome flotation assay, we found that after sucrose step centrifugation they did not float with liposomes and remained in the bottom fraction (*Figure 5d*). When we performed the same experiment without allowing the VLPs to recover, or when we only back-treated them for 5 or 15 min at 4°C, the 50°C-treated VLPs still floated with liposomes and were found in the upper fractions. Similarly, the sample still floated with liposomes when we back-treated for 1 h at 37°C instead of 4°C (*Figure 5d*). Together, these data revealed that the ANDV surface is at a thermodynamic equilibrium, which at the physiological temperature of 37°C dynamically fluctuates between closed and open forms of spikes.

We then examined the G187C Gc mutant VLPs in the same way. We found that 50% of the G187C mutant VLPs floated with the liposomes at 37°C, similar to wild type (*Figure 5e* and *Figure 5—figure supplement 1*). The inter-spike disulfide bond at the Gc:Gc interface therefore does not prevent the conformational equilibrium between closed and open forms of the spike, despite the higher Gn/Gc dissociation $T_m$ observed by BN-PAGE (*Figure 2b*). This result is in line with the Gc fusion loops being away from the Gc:Gc dimer contacts in the spikes (*Figure 1a*), and corroborates that the observed conformational transition is mainly an intra-spike effect.

## The open spike conformation is off-pathway in the acidic pH-triggered membrane fusion process

To assess whether the temperature-induced fusion loop exposure had an effect on viral infectivity, we incubated ANDV for 15 min at different temperatures ranging from 20°C to 56°C. Subsequently we infected cells through adsorption for 1 h at 37°C and quantified viral infection 16 h later. We found that the infection of cells by ANDV was strongly reduced depending on the temperature of the pre-treatment when the virus was adsorbed at 37°C (*Figure 6a*). But when after high temperature treatment the ANDV particles were allowed to recover the closed conformation of the spikes for 1 h at 4°C during adsorption to cells, their infectivity was completely restored, indicating that the observed inactivation is reversible (*Figure 6b*). Importantly, ANDV particles treated at 56°C did not recover their infectivity, in agreement with the reported temperature of 56°C required for hantavirus inactivation (*Kallio et al., 2006*).

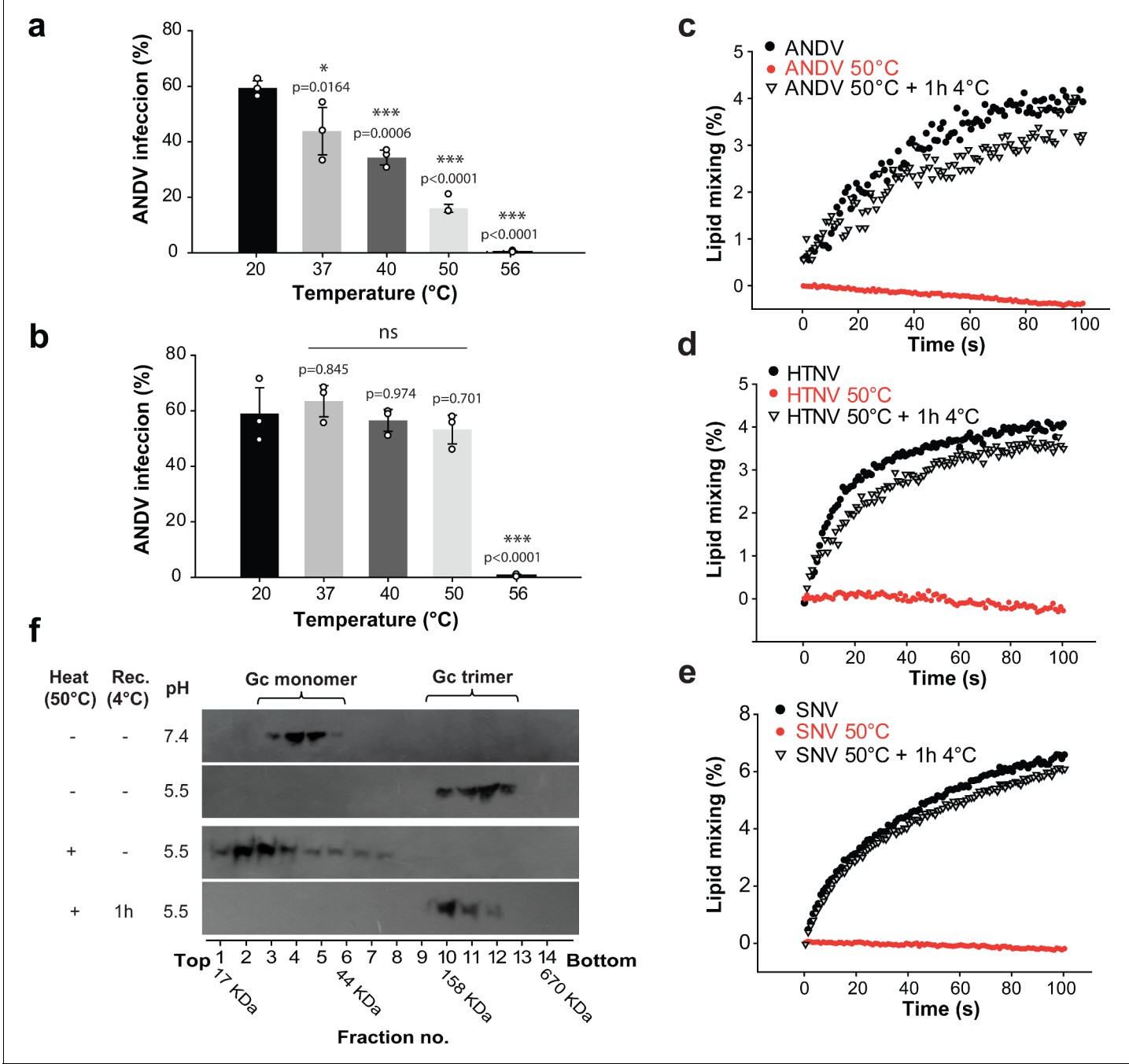

**Figure 6.** The open spike conformation does not induce viral infection and membrane fusion. (a-b) Infectivity of ANDV after temperature treatment and recovery at 37°C (a) or 4°C (b) during adsorption to cells for 1 h. Infection was quantified by flow cytometry 16 h later using anti-ANDV nucleoprotein antibody. Averages ± standard derivation from three biological replicates; ANDV treated at 20°C versus temperature-treated ANDV were analyzed by one-way ANOVA with a Bonferroni adjustment for multiple comparisons, p<0.001 (***), p<0.01 (**), p<0.05 (*) and non-significant (ns). (c-e) Lipid mixing of R18-labeled Andes VLPs (c), Hantaan virus VLPs (d) or Sin Nombre virus VLPs (e) with liposomes at pH 5.5. Before the VLPs were incubated with liposomes, they were either untreated, or incubated for 15 min at 50°C. Alternatively, the VLPs treated at 50°C were allowed to recover for 1 h at 4°C, before being subject to the lipid mixing assay. 'Rec' indicates recovery. R18 dequenching was continuously measured at 37°C and constant stirring. (f) Effect of heat-treatment of VLPs on acid-induced Gc trimerization. VLPs were either untreated, or incubated for 15 min at 50°C. Subsequently, they were incubated for 30 min at 37°C at the indicated pH. The Gc multimerization species were separated by sucrose gradient sedimentation and the fractions subjected to western blot using anti-Gc antibody. A molecular standard was used to estimate their molecular mass. Results of (c) to (f) are representative for three biological replicates.

DOI: https://doi.org/10.7554/eLife.46028.017

*Figure 6 continued on next page*

*Figure 6 continued*

The following source data is available for figure 6:

**Source data 1.** Data points for graphs and statistics of *Figure 6a–e*.
DOI: https://doi.org/10.7554/eLife.46028.018

To understand how the temperature-induced fusion loop exposure affects viral infectivity in mechanistic terms, we next analyzed the fusion activity of viral particles with liposomes. For this purpose, we labeled Andes VLPs with octadecyl rhodamine B (R18) and incubated them with liposomes. We observed R18 dequenching upon dropping the pH to 5.5, which indicated lipid mixing between the labeled VLPs and the unlabeled liposomes, as described previously (*Guardado-Calvo et al., 2016*). In turn, when we first incubated the Andes VLPs at 50°C and then mixed them with liposomes at acidic pH, we detected no lipid mixing, confirming the infectivity results obtained with authentic ANDV. We further tested whether the fusion activity could be restored when the 50°C-treated VLPs were incubated for 1 h at 4°C and then subjected to liposome fusion assays at acidic pH. Under these conditions, we found that the VLP-liposome fusion activity was restored (*Figure 6c*), again in line with the reversibility of the closed-to-open states transition and the virus infection results.

To test the validity of our observation across hantaviruses, we produced Sin Nombre VLPs as well as Hantaan VLPs and labeled them with R18. The labeled VLPs gave a clear lipid mixing signal upon acidification (*Figure 6d–e*), which was lost upon treatment at 50°C, as observed with Andes VLPs. When the 50°C-treated Sin Nombre or Hantaan VLPs were allowed to recover for 1 h at 4°C, they fully restored lipid mixing activity when incubated with liposomes at low pH (*Figure 6d–e*). Hence, these data suggest that the temperature-induced fusion loop exposure and reversible reduction of fusion activity is not an ANDV specificity but a property shared by hantaviruses in general.

To understand the molecular basis of the drop in viral infectivity and membrane fusion after treatment of hantavirus particles at high temperature, we tested whether the temperature treatment would still allow for Gc homo-trimerization towards the post-fusion form, as a measure of an early step in the virus-cell fusion mechanism. We thus incubated Andes VLPs at different temperatures and assayed them for low-pH induced Gc trimerization by sedimentation on a sucrose gradient. As expected, this method allowed the detection of Gc running as monomer at pH 7.4 and homotrimer at pH 5.5 when using untreated VLPs. But when we pre-treated the VLPs at 50°C and then incubated them at pH 5.5, we found Gc migrating predominantly as monomer. The temperature-treatment of the VLPs thus resulted in a form of Gc unable to undergo trimerization at acidic pH to induce membrane fusion (*Figure 6f*). When VLPs incubated at 50°C were back-treated for 1 h at 4°C and then incubated at pH 5.5, we found Gc again sedimenting as homotrimer, confirming the reversibility of the effect. This result indicates that in the open spike Gc is maintained in a form that cannot react to low pH by undergoing the fusogenic conformational change. Only when the spikes were allowed to adopt the closed conformation, they re-acquired the capacity to respond to low pH by allowing Gc homotrimerization to induce membrane fusion.

## Discussion

Here we have addressed the surface organization of pleomorphic hantavirus particles. By combining structural, biochemical and functional analyses, we revealed the molecular interface by which individual $(Gn/Gc)_4$ hetero-octameric spikes associate laterally via 2-fold related Gc:Gc contacts (*Figure 1*) akin to the contacts observed in alphavirus particles, and that an inter-spike disulfide bond across a 2-fold related Gc:Gc dimer interface improved the overall spike stability (*Figure 2b–c*). An assembly model in which 2-fold Gc contacts relate individual $(Gn/Gc)_4$ spikes is consistent with electron microscopy observations showing a continuous surface lattice of spikes that interact sidewise to form a grid-like pattern (*Battisti et al., 2011*; *Huiskonen et al., 2010*; *Li et al., 2016*; *Martin et al., 1985*) and with molecular assembly models proposed earlier (*Hepojoki et al., 2010*). Contrary to the study of *Hepojoki et al. (2010)* suggesting that Gc is mostly dimeric when solubilized from spikes, this and previous work show that Gc is monomeric when solubilized from viral particles (*Acuña et al., 2015*; *Barriga et al., 2016*) or when recombinantly expressed in the absence of the transmembrane segment (*Guardado-Calvo et al., 2016*; *Willensky et al., 2016*). These observations

are consistent with the small Gc:Gc contact patch, which appears too weak to maintain Gc dimers in solution. Similarly, membrane-anchored Gc expression in the absence of Gn was predominantly monomeric in live cells, although some Gc dimers were detected (*Sperber et al., 2019*). Therefore, it is likely that Gc anchoring in the context of lateral inter-spike constrains may be required for efficient Gc:Gc association. The residues involved in Gc:Gc contacts are highly conserved across hantaviruses in general, suggesting that our results can be extended to viruses across the mammal-infecting branch of the *Hantaviridae* family (*Figure 1—figure supplement 1*). Our results also show that mutation to residues that interfere with interface contacts significantly decrease virus particle production (*Figure 3a*), implying a role for these contact residues in viral assembly by connecting spikes laterally, building the viral surface lattices. Because hantaviruses do not have a matrix protein to induce membrane curvature, as do most of the other enveloped RNA viruses, the Gc:Gc contacts driving lateral interactions between spikes on the membrane are likely to play an important role in virion budding.

In spite of the unique four-fold symmetry of the hantavirus spikes, their overall arrangement on the particles has clear similarities to that of other class II enveloped viruses, such as alphaviruses, which display 3-fold symmetric spikes (*Lescar et al., 2001*). The inter-spike contact area of ~500 Å$^2$ of the 2-fold related hantavirus Gc:Gc (*Guardado-Calvo et al., 2016*) and alphavirus E1:E1 dimers (*Roussel et al., 2006*) include a highly conserved His residue at the center the dimer interface; Gc His303 and E1 His125. Their substitution decreases the pH threshold for acid-induced activation in hantaviruses (*Figure 4c* and *Figure 4—figure supplement 1c*) and in alphaviruses (*Qin et al., 2009*), suggesting in both cases that inter-spike dissociation is driven by repulsion upon protonation.

The results reported here have also revealed that the hantavirus spikes exhibit a dynamic equilibrium between closed and open forms, with the latter exposing the Gc fusion loops at physiological temperatures even at neutral pH (*Figure 5a*). We found that at 37°C - the physiological temperature of its rodent hosts – about 50% of the Andes VLPs bound to liposomes via Gc fusion loop exposure. The steep sigmoidal curve of liposome binding and detergent-solubilized spike dissociation as a function of temperature (*Figure 5c*) suggests a strongly cooperative effect. A possible explanation for these observations can be provided by assuming that the spikes change conformation intermittently, in a stochastic fashion, as represented in *Figure 7*.

Contrary to alphaviruses, hantavirus particle maturation does not involve proteolytic processing of the spikes during exocytosis. It is therefore possible that the observed reversibility of fusion loop exposure is related to the absence of irreversible proteolytic priming for fusion. Some mechanism must ensure, however, that Gc escapes from undergoing a premature irreversible conformational change triggered by the low pH environment of the exocytosis pathway of the cell, a mechanism that awaits to be discovered.

Our observation that hantavirus particles with spikes in the open form are not capable of inducing low-pH triggered membrane fusion (*Figure 6c–e*) correlates with the inability of Gc in these particles to form homotrimers (*Figure 6f*), suggesting that low pH treatment in the presence of multiple spikes in the open form engages Gc in non-functional interactions with itself from adjacent spikes, such that it cannot reach its trimeric post-fusion form (*Figure 7g*). There appears, however, to be a thin divide between the open conformation of the particles (around 50°C, *Figure 7c*) which allows recovery of their fusogenicity by returning to the ground state (i.e., a closed particle, *Figure 7b*), and an irreversible state in which they cannot recover it (i.e., treatment to 56°C) likely involving partial protein denaturation and aggregation (*Figures 5b* and *7e*).

*Figures 2a* and *5c* show that at room temperature the VLPs appear to be essentially in the closed form, where no flotation with liposomes is observed. This suggests that the stability of the infectious particle is much higher in the external environment, contributing to their propagation in nature. Within a mammalian host, there would be a much more rapid particle turnover. This notion is in line with previous data showing that hantaviruses are highly labile at 37°C, while displaying prolonged infectivity outside a host (*Kallio et al., 2006*). It is possible that the dynamic spike behavior at 37°C results from adaptation to their rodent hosts, providing advantages for establishing chronical infections; on the one hand the major lability of hantaviruses at 37°C may help to restrict their dissemination within the host by decreasing the time window for viral spread, while on the other hand the conformational diversity may represent an important decoy for escape from the hosts´ immune response. Furthermore, the observed dynamic behavior of the spikes is likely important for virus infectivity. In contrast to liposomes, the plasma membrane is a crowded environment with multiple

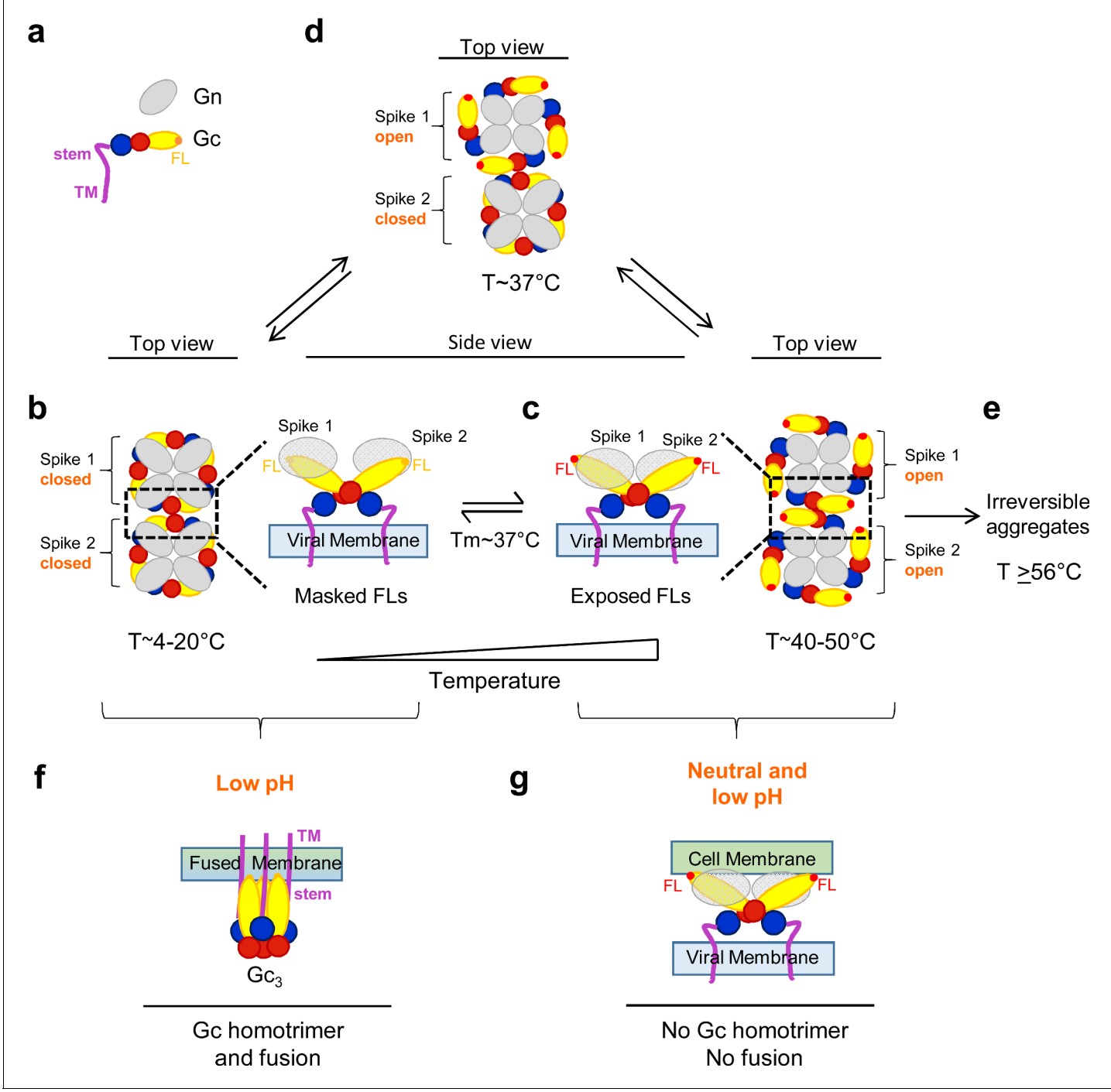

**Figure 7.** Hantavirus Gn/Gc spike dynamics and Gc conformational changes. Diagrammatic representation of 'breathing' spikes. (**a**) The Gn ectodomain is represented in gray and Gc in colors: red for domain I; yellow domain II; blue domain III; stem and transmembrane (TM) segment magenta. The Gc fusion loops are labeled FL. Gn-masked Gc FLs are in orange, while the exposed FLs are highlighted in red. (**b-d**) Top view of two (Gn/Gc)$_4$ hetero-octameric spikes on the viral surface. The spike 4-fold symmetry axis and the 2-fold axis relating the spikes are perpendicular to the plane of the Figure (radial in the particle). For clarity, only the Gn ectodomain is represented, and in the top views the Gc stem and TM segments are not drawn either. At the $T_m$ of 37°C (**d**), the Andes virus spikes are in equilibrium between closed (**b**) and open (**c**) forms, with Gc exposing the fusion loops in the latter. At lower temperatures, the equilibrium is shifted towards the closed conformation (**b**) and at higher temperatures toward the open conformation (**c**). The latter conformation is unstable, and leads to aggregates and inactivation at even higher temperatures (**e**). A side view closeup of two Gn/Gc heterodimers from the two spikes and related by Gc contacts at the inter-spike 2-fold symmetry axis is shown at the center, in between panels b) and c), corresponding to the area marked by dotted rectangles in the tops views. (**f**) Side view of a Gc homotrimer in the post-fusion conformation, bound

*Figure 7 continued on next page*

*Figure 7 continued*

to a fused membrane resulting from the low-pH triggered fusogenic conformational change of Gc. (**g**) Diagram represents an open spike interacting with membranes at neutral pH. Incubation at acidic pH in this conformation does not result in productive fusion, and no Gc homotrimer is formed.

DOI: https://doi.org/10.7554/eLife.46028.019

proteins and glycosaminoglycans, and whether fusion loop exposure can already allow for particle binding to cells is an open question. It could take place, for instance, after binding to a protein receptor (such as $\beta_3\alpha_V$ integrins) (*Choi et al., 2008*; *Gavrilovskaya et al., 1998*; *Jangra et al., 2018*) that could bring the hantavirus surface into proximity of a patch of naked membrane before its uptake by endocytosis. At any rate, at 37°C there should be enough spikes in the closed conformation on the virion to allow for productive fusion once in the acidic environment of the endosomes.

Similar conformational dynamics have been observed for a number of unrelated viruses (*Heyrana et al., 2016*; *Kuhn et al., 2015*; *Lewis et al., 1998*; *Munro et al., 2014*). It has been shown that a similar dynamic behavior of dengue virus particles elicits highly cross-reactive but poorly neutralizing antibodies targeting the conserved but cryptic fusion loop (*Beltramello et al., 2010*; *Oliphant et al., 2006*). These antibodies are believed to be responsible for antibody-dependent enhancement of the infection, which is the main obstacle to developing an efficient vaccine against dengue virus (*Rey et al., 2018*). The observed dynamic behavior of the hantavirus particles described here can have therefore an important impact in the development of suitable immunogens capable to confer protection against these pathogens (*Graham, 2017*; *Kotecha et al., 2015*; *McLellan et al., 2013*; *Rey and Lok, 2018*). Our data thus suggest the possibility of designing a subunit vaccine that exposes an inert 'closed spike' conformation only would elicit the strongest antibody response, similar to the closed form of the HIV Env trimers (*Torrents de la Peña et al., 2017*). Our results now pave the way for testing this type of approaches against hantavirus infections.

# Materials and methods

## Key resources table

| Reagent type (species) or resource | Designation | Source or reference | Identifiers | Add. inform. |
|---|---|---|---|---|
| Strain, strain background (*Andes virus*) | Andes virus isolate CHI-7913 | *Galeno et al., 2002* | | |
| Cell line (*Homo sapiens*) | 293FT | Thermo Fisher Scientific | Cat.:R700-07; RRID:CVCL_6911 | |
| Cell line (*Cercopithecus aethiops*) | Vero 76 (Vero E6) | American Type Culture Collection (ATCC) | CRL 1587; RRID:CVCL_0603 | |
| Recombinant DNA reagent | pI.18/GPC from ANDV | *Cifuentes-Muñoz et al., 2010* | | |
| Recombinant DNA reagent | pWRG/PUU-M(s2) | *Hooper et al., 2006* | | |
| Recombinant DNA reagent | pcDNA/Sin Nombre virus-GP | *Kleinfelter et al., 2015* | | |
| Biological sample | L-α-phosphatidylcholine (egg, chicken) | Avanti Polar Lipids | Cat.:840051C | |
| Biological sample | L-α-phosphatidyl-ethanolamine (egg, chicken) | Avanti Polar Lipids | Cat.:840021C | |
| Biological sample | Sphingomyelin (Brain, Porcine) | Avanti Polar Lipids | Cat.:860062C | |
| Biological sample | Cholesterol (ovine wool) | Avanti Polar Lipids | Cat.:700000P | |

*Continued on next page*

*Continued*

| Reagent type (species) or resource | Designation | Source or reference | Identifiers | Add. inform. |
|---|---|---|---|---|
| Biological sample | Gel Filtration Standard | Biorad | Cat.:1511901 | |
| Biological sample | NativeMark Unstained Protein Standard | Thermo Fisher Scientific | Cat.:LC0725 | |
| Biological sample | PageRuler Prestained Protein Ladder | Thermo Fisher Scientific | Cat.:26616 | |
| Antibody (monoclonal) | Mouse anti-ANDV nucleoprotein clone 7B3/F6 | *Tischler et al., 2008* | | 1:2000 dilution |
| Antibody (monoclonal) | Mouse anti-Gc clone 2H4/F6 | *Godoy et al., 2009* | | 1:2500 dilution |
| Antibody (monoclonal) | Mouse anti-Gn clone 6B9/F5 | *Cifuentes-Muñoz et al., 2010* | | 1:2500 dilution |
| Antibody (monoclonal) | Mouse anti-β-actin | Sigma | Cat:A2228; RRID:AB_476697 | 1:2500 dilution |
| Antibody (oligoclonal) | Goat anti-mouse immunoglobulin Alexa 555 conjugate | Thermo Fisher Scientific | Cat:A28180; RRID:AB_2536164 | 1:500 dilution |
| Antibody (polyclonal) | Goat anti-mouse IgG (H + L) horseradish peroxidase conjugate | Thermo Fisher Scientific | Cat:31430; RRID:AB_228307 | 1:5000 dilution |
| Antibody (oligoclonal) | Goat anti-mouse IgG (H + L) Alexa555 conjugate | Thermo Fisher Scientific | Cat:A28175; RRID:AB_2536161 | 1:500 dilution |
| Recombinant DNA reagent | Mutant constructs of pI.18/GPC | This paper, produced by GenScript, Piscataway, NJ. | | |
| Commercial assay or kit | Cell surface protein isolation kit | Pierce | Cat:89881 | |
| Commercial assay or kit | 1,6-diphenyl-1,3,5-hexatriene (DPH) | Sigma-Aldrich | Cat.:D208000 | |
| Commercial assay or kit | Octadecyl Rhodamine B Chloride (R18) | Thermo Fisher Scientific | Cat.:O246 | |
| Commercial assay or kit | 5-chloromethyl-fluorescein diacetate (Cell Tracker green CMFDA) | Thermo Fisher Scientific | Cat.:C7025 | |
| Commercial assay or kit | Lipofectamine 2000 | Thermo Fisher Scientific | Cat.:11668019 | |
| Software, algorithm | GraphPad Prism, version 6, and SPSS software (SPSS, Inc) | GraphPad Software | | |
| Software, algorithm | SigmaPlot 12.0 | Systat Software | | |
| Other (*Hantaan virus*) | Gc ectodomain structure from Hantaan virus | *Guardado-Calvo et al., 2016* | PDB: 5LJY | |
| Other (Semliki Forest *virus*) | E1 ectodomain structure from Semliki Forest virus | *Roussel et al., 2006* | PDB: 2ALA | |

## Virus and cells

ANDV isolate CHI-7913 (*Galeno et al., 2002*) (kindly provided by Héctor Galeno, Instituto de Salud Pública, Chile) was propagated in Vero E6 cells (ATCC) as described before (*Barriga et al., 2013*). All work involving the infectious ANDV was performed under strict biosafety level three conditions (Centro de Investigaciones Médicas, Pontificia Universidad Católica de Chile, Chile). 293FT cells (Thermo Fisher Scientific) were propagated in DMEM supplemented with 10% fetal calf serum (FCS). Vero E6 cells (ATCC) were grown in MEM containing 10% FCS, non-essential amino acids and 1 mM

sodium pyruvate (Thermo Fisher Scientific). STR profiling was performed for human cell line authentication (ATCC) and mycoplasma testing was negative for all used cell lines.

## Expression plasmids and design of Gc mutants

For ANDV Gn/Gc expression we used the plasmid pI.18/GPC that codes for the full length GPC of ANDV strain CHI-7913 under the control of the cytomegalovirus promotor (*Cifuentes-Muñoz et al., 2010*). Site-directed mutations were generated by DNA synthesis and sub-cloning into pI.18/GPC using intrinsic restriction sites (GenScript). To express the full length GPC from Hantaan virus or Sin Nombre virus, the plasmids pWRG/HTN-M(x) (*Hooper et al., 2006*) (kindly donated by Dr. Jay Hooper, USMARIID) and pcDNA/Sin Nombre virus-GP plasmid (*Kleinfelter et al., 2015*) were used (kindly provided by Drs. Kartik Chandran and Rohit Jangra from Albert Einstein College of Medicine).

## Expression of Gn/Gc and assembly into VLPs

For Gn/Gc expression, 293FT cells (Thermo Fisher Scientific) were grown in 100 mm plates and calcium-transfected with the corresponding GPC encoding plasmid. 48 h later, cell surface proteins were labeled with biotin in order to separate the biotinylated (surface proteins) from non-biotinylated (intracellular proteins) fractions using a cell surface protein isolation kit (Pierce). For protein detection by western blot, primary anti-Gc monoclonal antibody (MAb) 2H4/F6 (*Godoy et al., 2009*), anti-Gn monoclonal antibody 6B9/F5 (*Cifuentes-Muñoz et al., 2010*) or anti-β-actin MAb (Sigma) were used at 1:2500 and subsequently detected with an anti-mouse immunoglobulin horseradish peroxidase conjugate (Thermo Fisher Scientific) 1:5000 and a chemiluminescent substrate (Pierce). All these antibodies were previously characterized concerning their reactivity with negative controls (*Cifuentes-Muñoz et al., 2011*; *Cifuentes-Muñoz et al., 2010*). VLPs were harvested from supernatants of 293FT cells transfected with the pI.18/GPC wild type or the different mutant constructs at 48 h post-transfection and concentrated as previously established (*Acuña et al., 2014*). The amount of GPC encoding plasmid used for each mutant was adjusted in order to reach similar amounts of cell surface accumulation to wild type.

## BN-PAGE and western blotting

The discontinuous native protein gel electrophoresis was performed similar to as previously described (*Niepmann and Zheng, 2006*). Briefly, VLP samples harvested from pI.18/GPC wild type or the different mutant constructs were incubated with Coomassie G-250 0.25% and Triton X-100 0.5% for 15 min at different temperatures just before loading onto a 3–16% gradient polyacrylamide gel. The native gel electrophoresis was run at 130 mV for 15 h at 4°C. The buffering system was 200 mM Tris for anode buffer and 50 mM Tris, 100 mM glycine cathode buffer. The size of Gn and Gc species was estimated using the migration rate of a molecular standard (Native Mark Unstained Protein Standard, Invitrogen) which was independently stained. The rest of the gel was incubated in transfer buffer at room temperature. After the transfer, the nitrocellulose was blocked in PBS including 5% skim milk and next Gn and Gc glycoproteins were stained separately by using anti-Gn MAb 6B9/F5 (*Cifuentes-Muñoz et al., 2010*) and anti-Gc MAb 2H4/F6 or 5D11/G7 (*Godoy et al., 2009*) at a 1:2500 dilution each. Primary antibody staining was detected as described above. For the quantification of the Gn/Gc spike dissociation, the densitometry values of dissociated Gc (monomeric or dimeric Gc) were divided by the densitometry values of the total signal for Gc, using the ImageJ software (*Schneider et al., 2012*). The average value and standard derivation (s.d.) of biological replicates was calculated for each temperature condition and a sigmoidal curve fitted using *Equation 1*.

$$\text{Gn/Gc dissociation (\%)} = \text{Gn/Gc disociation (20°C)} + \text{Gn/Gc dissociation MAX}/\left(1 + e^{-(T-Tm)/b}\right) \quad (1)$$

were Gn/Gc dissociation (20°C) is the basal dissociation at 20°C, Gn/Gc dissociation *MAX* is the maximal dissociation value, $T_m$ is the temperature at 50% Gn/Gc dissociation and *b* is the Hill's slope of the curve, indicating its steepness. The curve was fitted using a sigmoidal four parameters equation in SigmaPlot 12.0, Systat Software.

## VLP-liposome coflotation

Liposomes were prepared fresh by the freeze-thaw and extrusion method (*Castile and Taylor, 1999*). PC (phosphatidylcholine, from chicken egg) and PE (phosphatidylethanolamine, from chicken egg), sphingomyelin (from porcine brain), cholesterol (from ovine wool), were purchased from Avanti Polar Lipids and large multilaminar vesicles (liposomes) were prepared using PC/PE/sphingomyelin/cholesterol in a 1/1/1/1.5 ratio respectively. The coflotation of viral particles with liposomes was performed as previously established (*Acuña et al., 2015*). First, liposomes were labeled with 200 mM 1,6-diphenyl-1,3,5-hexatriene (DPH) and VLPs prepared from wild type or mutant pl.18/GPC constructs were incubated at pH 5.5 for 15 min at 37°C as positive control or at pH 7.4 using different temperatures. The VLP-liposome mixture was then added to the bottom and adjusted to 25% (w/v) sucrose. Additional sucrose steps of 15% and 5% were then over-layered. After centrifugation for 2 h at 300,000 g, liposomes were detected by the fluorescence emission of DPH ($\lambda$ex = 230 nm; $\lambda$em = 320 nm) and VLPs by western blot using anti-Gc MAb 2H4/F6.

The western blot signal was quantified using the ImageJ software (*Schneider et al., 2012*) and VLP-liposome coflotation calculated by dividing the densitometry value of liposome associated Gc by the densitometry value of the total signal for Gc. The average value and s.d. of biological replicates was calculated and a sigmoidal VLP-liposome coflotation curve fitted using *Equation 2*.

$$\text{VLP coflotation}\,(\%) = \text{VLP coflotation}\,(20°C) + \text{VLP coflotation MAX}*\left(1 + e^{-(T-Tm)/b}\right) \qquad (2)$$

were VLP coflotation (20°C) is the VLP coflotation at 20°C, VLP coflotation *MAX* is the maximal VLP coflotation percentage, *Tm* is the temperature at 50% VLP coflotation and *b is* the Hill's slope of the curve. The curve was fitted using a sigmoidal four parameters equation in SigmaPlot 12.0, Systat Software.

## Infection of cells with ANDV

ANDV was incubated at different temperatures for 15 min and subsequently added to Vero E6 cells (MOI = 1). After 1 h of adsorption at 4°C or 37°C, the cells were washed in excess and next infection was allowed to proceed for 16 h by incubation in MEM 10% FBS at 37°C. Quantification of viral infection was performed as previously described (*Barriga et al., 2013*) by detecting viral nucleoprotein expressing cells by using flow cytometry. Briefly, infected cells were detached and fixed with 2% paraformaldehyde for virus inactivation. Subsequently, the fixed cells were permeabilized using 0.1% Triton X-100 and then stained with primary MAb 7B3/F6 anti-ANDV nucleoprotein (*Tischler et al., 2008*) by incubation for 2 h at RT, which in turn was detected by goat anti-mouse immunoglobulin conjugated to Alexa Fluor 488 (Thermo Fisher Scientific). Flow cytometry was performed in a cytometer (FACS CAN II, Becton Dickinson) counting at least 5,000 cells. The gate for ANDV nucleoprotein positive cells was established using as negative control non-infected cells labeled with the same primary and secondary antibodies.

## VLP-liposome lipid mixing

For the lipid mixing assay, VLPs were labeled with 1 µg/ml of R18 (Invitrogen). Labeled VLPs were then mixed with liposomes in a continuously stirred fluorimeter cuvette at 37°C and lipid mixing was monitored by the decrease in R18 fluorescence generated by the dilution of the R18 probe with the unlabeled phospholipids in the liposome membrane. Fluorescence was recorded continuously at 580 nm using a fluorescence spectrophotometer (Varian Eclypse, Agilent Technologies) at an excitation wavelength of 560 nm using a 10 nm slit width for excitation and emission. After a stable base line was established at pH 7, it was subtracted from recording and established as base line value corresponding to 0% lipid mixing. The reaction initiation time (t = 0) corresponds to the lowering of the pH to 5.5. The maximal extent (100%) of excimer dilution was defined by the addition of Triton X-100 0.1% (v/v) after lipid mixing of each condition had concluded.

## Sucrose gradient sedimentation

Acid-induced Gc homotrimerization was tested as established before (*Acuña et al., 2015*). VLPs constituted of ANDV Gn/Gc were incubated for 30 min at 37°C at the indicated pH to induce multimerization changes. Next, Triton X-100 1% (v/v) was added to allow the extraction of the membrane

glycoproteins from the viral particle. The extracted glycoproteins were then added to the top of a sucrose gradient (7–15%; w/v) and centrifuged at 150,000 g for 16 h. Next, fractions were collected and the presence of Gc was analyzed by western blot using MAbs anti-Gc 2H4/F6. The molecular mass of fractions was determined experimentally by a molecular marker (Biorad).

## Cell-cell fusion

This three-color fluorescence assay was performed as previously described (*Cifuentes-Muñoz et al., 2011*). Vero E6 cells (ATCC) seeded into 16 well chamber slides were transfected with the pI.18/GPC wild type or the different mutant constructs using lipofectamine 2000 (Thermo Fisher Scientific). The DNA amounts were adjusted to obtain similar levels of Gc at the cell surface. 48 h later, the cells were incubated in E-MEM (pH 5.5) at 37°C for 5 min, subsequently washed with PBS, and the incubation continued for 3 h at 37°C in E-MEM (pH 7.2). The cell cytoplasm was then stained for one hour with 1 µM of 5-chloromethylfluorescein diacetate (Cell Tracker green CMFDA, Thermo Fisher Scientific) and cells then fixed for 20 min with 4% paraformaldehyde. For immunelabelling, the cells were then permeabilized with PBS 0.1% Triton X-100 and Gc stained using the monoclonal antibody 2H4/F6 1:500 and secondary antibody goat anti-mouse immunoglobulin conjugated to Alexa Fluor555 1:500 (Thermo Fisher Scientific). Finally, nuclei were stained for 5 min with DAPI 1 ng/µL and samples examined under a fluorescence microscope (BMAX51, Olympus). The fusion index of Gc expressing cells was calculated using the formula: 1- [number of cells/number of nuclei]. Approximately 200 nuclei per field were counted (10X magnification) and five fields used to calculate the fusion index for each sample of at least three biological replicates.

## Statistic analysis

All statistical analyses were carried in GraphPad Prism, version 6, and SPSS software (SPSS, Inc).

## Molecular graphics and structure analyses

For protein structure analyses and graphics PyMOL Molecular Graphics System Version 2.0 (Schrödinger, LLC) was used.

# Acknowledgements

We thank the use of the BSL3 facility of the Centro de Investigaciones Médicas (CIM), Facultad de Medicina de la Pontificia Universidad Católica de Chile. We also thank Dr. Héctor Galeno from Instituto de Salud Pública de Chile for providing Andes virus CHI-7913, Dr. Jay Hooper from USAMRIID for providing the plasmid pWRG/PUU-M(s2) and Drs. Kartik Chandran and Rohit Jangra from Albert Einstein College of Medicine for providing the pcDNA/Sin Nombre virus-GP plasmid.

# Additional information

## Competing interests

Eduardo A Bignon, Pablo Guardado-Calvo, Félix A Rey, Nicole D Tischler: Is named inventor on a patent application describing disulfide bonds for hantavirus spike stabilization (PCT/US19/22134). The other author declares that no competing interests exist.

## Funding

| Funder | Grant reference number | Author |
| --- | --- | --- |
| Comisión Nacional de Investigación Científica y Tecnológica | Fondo Nacional de Desarrollo Científico y Tecnológico FONDECYT 1181799 | Nicole D Tischler |
| Comisión Nacional de Investigación Científica y Tecnológica | Programa de Apoyo a Centros con Financiamiento Basal 170004 to Fundación Ciencia and Vida | Nicole D Tischler |

| Comisión Nacional de Investigación Científica y Tecnológica | FONDEQUIP EQM130092 for the improvement of BSL3 of Pontificia Universidad Católica de Chile | Nicole D Tischler |
| --- | --- | --- |
| Labex IBEID | ANR-10-LABX-62-IBEID | Félix A Rey |
| Labex IBEID | ANR-10-LABX-62-IBEID 4E AAP | Pablo Guardado-Calvo Félix A Rey |
| Seventh Framework Programme | Infect-ERA IMI European network HantaHunt Program | Pablo Guardado-Calvo Félix A Rey |
| Comisión Nacional de Investigación Científica y Tecnológica | Fondo Nacional de Desarrollo Científico y Tecnológico FONDECYT 3150695 | Amelina Albornoz |

The funders had no role in study design, data collection and interpretation, or the decision to submit the work for publication.

### Author contributions

Eduardo A Bignon, Conceptualization, Data curation, Software, Formal analysis, Validation, Investigation, Visualization, Methodology, Writing—original draft, Writing—review and editing; Amelina Albornoz, Resources, Data curation, Investigation, Methodology; Pablo Guardado-Calvo, Conceptualization, Writing—review and editing; Félix A Rey, Conceptualization, Formal analysis, Funding acquisition, Writing—review and editing; Nicole D Tischler, Conceptualization, Resources, Formal analysis, Supervision, Funding acquisition, Validation, Visualization, Methodology, Writing—original draft, Project administration, Writing—review and editing

### Author ORCIDs

Eduardo A Bignon https://orcid.org/0000-0002-9116-3352
Amelina Albornoz https://orcid.org/0000-0003-1030-8650
Pablo Guardado-Calvo https://orcid.org/0000-0001-7292-5270
Félix A Rey https://orcid.org/0000-0002-9953-7988
Nicole D Tischler https://orcid.org/0000-0002-4578-4780

### Decision letter and Author response

Decision letter https://doi.org/10.7554/eLife.46028.022
Author response https://doi.org/10.7554/eLife.46028.023

## Additional files

### Supplementary files

• Transparent reporting form
DOI: https://doi.org/10.7554/eLife.46028.020

### Data availability

All data generated or analysed during this study are represented in the manuscript. Numerical data and statistics summary data source is provided for all graphs (Figures 2C, 3A, 3B, 4A, 4B, 4C, 5C, 5E, 6A, 6B, 6C, 6D and 6E).

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
