## [Decision Letter]

Thank you for submitting your article "Molecular organization and dynamics of the fusion protein Gc at the hantavirus surface" for consideration by *eLife*. Your article has been reviewed by Neil Ferguson as the Senior Editor, a Reviewing Editor, and three reviewers. The reviewers have opted to remain anonymous.

The reviewers have discussed the reviews with one another and the Reviewing Editor has drafted this decision to help you prepare a revised submission.

Summary:

Guided by structural information, Bignon and colleagues subjected the Andes virus (ANDV) Gn/Gc spike to mutagenic analysis in order to identify Gc residues important for contact between spikes. They show that the interface between two Gc proteins in their pre-fusion form, as detected previously by crystallography, is the same one that mediates Gc:Gc contacts between adjacent spikes on the virion surface. Specifically, they demonstrate that disulfide bonds can be formed between certain residues located at this interface and that an intact interface is required for assembly of virus-like particles. Moreover, residues located at the interface are shown to affect spike stability and responsiveness to low pH. Finally, they demonstrate that the fusion loop in Gc can be reversibly exposed at elevated temperatures – the spike transits between an open and a closed form – and that the open form is not compatible with low pH induced membrane fusion activation.

Essential revisions:

All three referees agree that the study is of interest and that the study provides reasonably solid experimental support for the hypothesis that the Gc:Gc intermolecular contacts observed in the crystal structure of Hantaan virus Gc represent physiologically relevant intermolecular contacts that contribute to the assembly and stability of hantavirus particles. Moreover, the observations that thermal destabilization of Andes virus or VLPs from other hantaviruses causes a conformational change that exposes the fusion loop and inhibits low pH-dependent cell entry will be interest to researchers studying hantaviruses and potential treatments or vaccines against them. However, for the study to be acceptable the following points need to be addressed:

1) The authors claim that the Gc:Gc dimer interface is "required for VLP assembly" (subsection “The inter-spike Gc dimer contacts are required for VLP assembly”). However, in Figure 3, seven of the eleven Gc:Gc interface mutants tested positive for Gc in the "VLP" cell supernatant fraction, indicating that these mutants can still form VLPs. When corrected for actin expression, the expression of Gc in these mutants does not appear significantly impaired, contrary to the authors' claim. Thus, in order to substantiate the author's conclusions, immunoblots should be quantified and averages +/- SD should be presented. Additionally, of the four mutants that failed to generate VLPs, three showed no Gc expression at all, indicating that the mutations prevented Gc expression and/or folding rather than VLP assembly per se, as claimed in subsection “The inter-spike Gc dimer contacts are required for VLP assembly”. Only one mutant, H303E, was expressed in cells but did not produce VLPs. Hence the claim that "Gc dimer contacts are required for VLP assembly" is unconvincing and not supported by the data presented. The authors should remove this claim (and related claims) perhaps replacing it with "…contacts are required for Gn/Gc assembly, or glycoprotein spike assembly".

Alternatively, the authors would need to provide additional evidence to support their original claims, such as one or more pairs of interacting charge reversal mutations that rescue VLP assembly whereas each individual charge reversal mutation from one any one pair is deficient for VLP assembly but not for Gc expression in cells.

2) Discussion of the results and representation of the literature:

In the Introduction the authors quite sparsely cite studies by other groups, and in the Discussion previous knowledge and hypotheses by other groups are not appropriately considered. For instance, the authors do not discuss the findings presented in Hepojoki et al. (2010), especially since it contains studies on covalent bonding between hantavirus glycoproteins. In addition to the work by other groups mentioned in the introduction, the authors may consider a recent paper by Sperber et al. (2019). Also, it is unclear why the authors did not present Gc docked into the 3D structure of the spike complex (which is available from database). Does this mean that the Gc crystal structure does not fit into the spike structure, and if so, is the orientation of the domains in respect to each other as the author claim? Generally, the authors may expand the Discussion to put the results in perspective regarding previous studies and focus more on hantaviruses.

---

## [Author Response]

Essential revisions:1) The authors claim that the Gc:Gc dimer interface is "required for VLP assembly" (subsection “The inter-spike Gc dimer contacts are required for VLP assembly”). However, in Figure 3, seven of the eleven Gc:Gc interface mutants tested positive for Gc in the "VLP" cell supernatant fraction, indicating that these mutants can still form VLPs. When corrected for actin expression, the expression of Gc in these mutants does not appear significantly impaired, contrary to the authors' claim. Thus, in order to substantiate the author's conclusions, immunoblots should be quantified and averages +/- SD should be presented.

We apologize for this omission and thank the referee for their helpful suggestions. We have now included the quantification of the immunoblots +/- SD for VLP release into the cell supernatant including statistical analyses (revised Figure 3A). In this figure, the amounts of VLP release of each mutant can now be compared with the corresponding signal for wild type VLPs. To ensure equal amounts of properly folded GPC for VLP release, the amount of cell surface accumulation equivalent to wild type GPC was adjusted by the amount of GPC encoding plasmid used for transfection. The corresponding western blots have been moved to Figure 3—figure supplement 1. As can be seen, each experiment includes different mutants together with the wild type as a control (used to normalize the signal for VLP release to 100%). A significant decrease of VLP assembly can be observed in the case of the alanine mutant D28A and opposite charge mutants (revised Figure 3A) while maintaining their trafficking to the plasma membrane (Figure 3—figure supplement 1C and D). In the cellular fraction, equal amounts of proteins in of cellular fraction loaded in each lane of the western blots were normalized by total protein quantification. Anti-β-actin was only used to detect plasma membrane integrity, as cytosolic protein control. All corresponding modifications have been introduced throughout the text (Results section and Materials and methods section) and the corresponding figure legends.

Additionally, of the four mutants that failed to generate VLPs, three showed no Gc expression at all, indicating that the mutations prevented Gc expression and/or folding rather than VLP assembly per se, as claimed in subsection “The inter-spike Gc dimer contacts are required for VLP assembly”. Only one mutant, H303E, was expressed in cells but did not produce VLPs.

This is correct. To ease the interpretation of the results, in the revised Figure 3A we now only show VLP release of those mutants that were properly expressed and reached the plasma membrane. The original western blots indicating expression/folding and VLP release of all mutants have been moved to Figure 3—figure supplement 1A-C.

Hence the claim that "Gc dimer contacts are required for VLP assembly" is unconvincing and not supported by the data presented. The authors should remove this claim (and related claims) perhaps replacing it with "…contacts are required for Gn/Gc assembly, or glycoprotein spike assembly".

We respectfully disagree here with the reviewer, as we are not claiming that the Gc dimer interface is required for Gn/Gc spike assembly, but rather for formation of the surface lattice that makes the VLP, via inter-spike contacts. But the reviewer’s remark make us realize that we were not clear enough in the original version of the manuscript. Our data clearly shows that substitutions by alanine or by residues bearing opposite charges, significantly decreases VLP assembly compared to wild type (please see graph of revised Figure 3A). Based on the helpful reviewer’s suggestions to quantify our data, and to remove mutants from the analysis that are not properly expressed, we believe that Figure 3A now is clearer in showing that “inter-spike Gc dimer contacts are required for VLP assembly”. The revised text was accordingly adjusted, explaining the results better.

Alternatively, the authors would need to provide additional evidence to support their original claims, such as one or more pairs of interacting charge reversal mutations that rescue VLP assembly whereas each individual charge reversal mutation from one any one pair is deficient for VLP assembly but not for Gc expression in cells.

We agree with the reviewer that charge reversal of pairs of interacting residues rescuing VLP formation would be a nice validation of our model. However, such experiments would take a lot of time, and we have shown here that engineering a cysteine residue at position G187 leads to disulfide crosslinked spikes, as expected if the dimer interface displayed in Figure 1B is the one relating two spikes. In our view, the disulfide bond is already a very strong evidence, and the additional mutagenesis only corroborates this result, by indicating that mutation of the other residues displayed in Figure 1B impacts the ease with which VLPs are formed. In other words, we agree the results reported in this session (Figure 3A) alone are not convincing evidence that this is the dimer interface, but in combination to the disulfide bond explained in the previous section (Figure 1 and Figure 2), they are. We therefore believe that the additional experiments suggested are not necessary.

2) Discussion of the results and representation of the literature:In the Introduction the authors quite sparsely cite studies by other groups, and in the Discussion previous knowledge and hypotheses by other groups are not appropriately considered. For instance, the authors do not discuss the findings presented in Hepojoki et al. (2010), especially since it contains studies on covalent bonding between hantavirus glycoproteins.

We had cited the work of Hepojoki et al. (2010) in the Introduction of the original submission. To address this comment, we have now better highlighted the major findings of that paper in the revised Introduction:

“The work of Hepojoki et al. (2010) revealed that Gn/Gc species can be covalently crosslinked on the surface of virions and suggested oligomeric models for spike assembly based on the characterization of detergent-solubilized spikes.”

In addition to the work by other groups mentioned in the introduction, the authors may consider a recent paper by Sperber et al. (2019).

We have now included this very recent paper in the Introduction of our manuscript as follows:

“In this line, a central role of Gn in self-association has recently been confirmed by number and brightness analysis in live, single cells showing that separate Gn expression allows detection of Gn oligomers while separate Gc expression predominantly leads to Gc monomers and some Gc dimers (Sperber et al., 2019).”

Also, it is unclear why the authors did not present Gc docked into the 3D structure of the spike complex (which is available from database). Does this mean that the Gc crystal structure does not fit into the spike structure, and if so, is the orientation of the domains in respect to each other as the author claim?

We thank the reviewers for raising this issue. In our paper, we limit ourselves to the functional validation of the data observed in the crystallographic structure. Although crystal packing contacts have to be taken with caution, our data are further supported by the structural homology of Gc with alphavirus spikes, which interact in a similar way via the glycoprotein E1. Crystal contacts have also been shown to be valid in other cases, such as the Ebola virus matrix protein, where contacts about a crystallographic 2-fold axis were later found to be indeed involved in particle assembly (Bornholdt et al., 2013).

Generally, the authors may expand the Discussion to put the results in perspective regarding previous studies and focus more on hantaviruses.

We now included additional discussion of our results concerning the biochemical approach in the field (Hepojoki et al., 2010) and hantavirus infectivity in dependence of time and temperature (Kallio et al., 2006). The discussion also includes comparison with structural information (Battisti et al., 2011; Guardado-Calvo et al., 2016; Husikonen et al., 2011; Martin et al., 1985; Shi et al., 2016; Willensky et al., 2016), functional approaches (Acuña et al., 2015; Barriga et al., 2016) and cell entry studies (Choi et al., 2008; Gavrilovskaya et al., 1998; Jangra et al., 2018). We believe that this literature comprises the most relevant previous hantavirus studies related to this work. Given the similarity of hantaviruses to other viruses with class II fusion proteins or with viruses with a highly dynamic surface behavior, the comparison with those additional models is properly justified and helps drawing of future directions.